# Comparative Efficacy and Safety of Immunotherapeutic Regimens with PD-1/PD-L1 Inhibitors for Previously Untreated Extensive-Stage Small Cell Lung Cancer: A Systematic Review and Network Meta-Analysis

Koichi Ando [1,2,*], Ryo Manabe [1], Yasunari Kishino [1], Sojiro Kusumoto [1], Toshimitsu Yamaoka [3], Akihiko Tanaka [1], Tohru Ohmori [1], Tsukasa Ohnishi [1] and Hironori Sagara [1]

[1] Division of Respiratory Medicine and Allergology, Department of Medicine, Showa University School of Medicine, 1-5-8 Hatanodai, Shinagawa-ku, Tokyo 142-8666, Japan; r.manabe@med.showa-u.ac.jp (R.M.); ookiyookiy@med.showa-u.ac.jp (Y.K.); k-sojiro@med.showa-u.ac.jp (S.K.); tanakaa@med.showa-u.ac.jp (A.T.); ohmorit@med.showa-u.ac.jp (T.O.); tohnishi@med.showa-u.ac.jp (T.O.); sagarah@med.showa-u.ac.jp (H.S.)
[2] Division of Internal Medicine, Showa University Dental Hospital Medical Clinic, Showa University Senzoku Campus, 2-1-1 Kita-Senzoku, Ohta-ku, Tokyo 145-8515, Japan
[3] Advanced Cancer Translational Research Institute (Formerly Institute of Molecular Oncology), Showa University, 1-5-8 Hatanodai, Shinagawa-ku, Tokyo 142-8555, Japan; yamaoka.t@med.showa-u.ac.jp
[*] Correspondence: koichi-a@med.showa-u.ac.jp; Tel.: +81-3-3784-8532; Fax: +81-3-3784-8742

**Abstract:** Improving therapeutic strategies for extensive-stage small cell lung cancer (ES-SCLC) remains a challenge. To date, no reports have directly compared the efficacy and safety of immune checkpoint inhibitors plus platinum–etoposide (ICIs+EP) with platinum–irinotecan (IP) or directly compared different ICIs+EP for previously untreated ES-SCLC. This study used a Bayesian approach for network meta-analysis to compare efficacy and safety between ICIs+EP and IP and between each pair of three ICIs+EP. The six treatment arms were: pembrolizumab plus platinum–etoposide (Pem+EP), durvalumab plus platinum–etoposide (Dur+EP), atezolizumab plus platinum–etoposide (Atz+EP), platinum–amrubicin (AP), IP, and platinum–etoposide (EP). No significant differences in overall survival were observed between ICIs+EP and IP and between each pair of three ICIs+EP. The incidence of ≥grade 3 adverse events (G3-AEs) was significantly higher in ICIs+EP than IP, whereas no significant difference was found in G3-AEs between each pair of three ICIs+EP. The incidence of ≥grade 3 neutropenia and thrombocytopenia was significantly higher in ICIs+EP than IP, whereas the incidence of ≥grade 3 diarrhea was significantly lower in ICIs+EP than IP. These findings will help clinicians better select treatment strategies for ES-SCLC.

**Keywords:** small cell lung cancer; immune checkpoint inhibitor; systematic review; network meta-analysis

## 1. Introduction

Lung cancer is responsible for most cancer-induced mortality worldwide, with small cell lung cancer (SCLC) accounting for 15% of newly diagnosed lung cancer cases [1–3]. The prognosis is unfavorable for extensive-stage SCLC (ES-SCLC), which accounts for 80% to 85% of newly diagnosed SCLC, with a reported median survival time of 7 to 10 months and a 5-year survival rate of not more than 8%, despite a systemic cancer treatment response rate of more than 50% [1–5]. Therefore, new treatment strategies for ES-SCLC are urgently required [2–4].

Recently, the efficacy of immune checkpoint inhibitors (ICIs) such as Programmed Cell Death-1 (PD-1)/Programmed Death-Ligand 1 (PD-L1) inhibitors (pembrolizumab (Pem), durvalumab (Dur), or atezolizumab (Atz)) plus platinum–etoposide (EP) for previously untreated ES-SCLC has been reported [6–8]. Compared with EP treatment alone, Pem+EP,

Dur+EP, or Atz+EP treatment has been shown to significantly improve overall survival (OS) and progression-free survival (PFS) [6–8], highlighting them as new treatment options for previously untreated ES-SCLC [3,9,10]. In North America, based on the results of the SWOG S0124 study [11], (platinum–irinotecan) IP is not as commonly used as EP, whereas in Japan, IP is currently one of the leading treatment options for previously untreated ES-SCLC based on the results of previous phase III studies and meta-analysis [12–15].

However, to date, no randomized study has compared ICIs+EP (Pem+EP, Dur+EP, or Atz+EP) with IP. Therefore, the efficacy and safety profile of ICIs+EP compared with that of IP for previously untreated ES-SCLC has not been fully evaluated. Although ICIs+EP is the available treatment option for previously untreated ES-SCLC [6–8], there is currently limited justification for selecting ICIs+EP over the other existing regimens, including IP. Furthermore, to date, there are no reports of randomized controlled trials (RCTs) that directly compare the efficacy and safety between Pem+EP, Dur+EP, and Atz+EP regimens. Therefore, there is little basis for oncologists to choose the most appropriate regimen among these immunotherapeutic regimens.

Direct head-to-head RCTs are primarily used for comparing the efficacy and safety of drugs. However, RCT studies are labor-intensive, time-consuming, and expensive. Therefore, we aimed to conduct a network meta-analysis (NMA) using data mined from the literature [16] to enable comparisons among pairs of treatments in the absence of existing head-to-head RCTs and rank the efficacy and safety of each therapeutic regimen using less time, effort, and expense [16–24]. In this systematic review of data (registered UMIN-CTR number: UMIN000041702), we compared and ranked the efficacy and safety of Pem+EP, Dur+EP, Atz+EP, platinum–amrubicin (AP), IP, and EP as first-line treatments for patients with previously untreated ES-SCLC using the statistical Bayesian NMA method. This not only facilitates a comparison between ICIs+EP and IP but also allows for a comparison of all treatment groups, including the ICIs (Pem, Dur, or Atz)+EP treatment groups. Our findings may provide oncologists with valuable information to better select therapeutic strategies for patients and develop new treatment strategies.

## 2. Materials and Methods

### 2.1. Systematic Review

A global literature review of four databases (PubMed [25], Embase [26], CENTRAL [27], and SCOPUS [28]) was performed in December 2020 to identify RCTs assessing ICIs+EP (Pem+EP, Dur+EP, or Atz+EP), IP, AP, or EP treatments for advanced SCLC, particularly ES-SCLC, published from 1 January 1946 onwards. No restrictions were placed on the publication date in the electronic database used for the literature search, other than the publication date had to be after 1 January 1946. The search strategy was selected using keywords such as "extensive stage", "small cell lung cancer", "immune checkpoint inhibitor", "pembrolizumab", "durvalumab", "atezolizumab", "irinotecan", and "etoposide", and their Medical Subject Headings (MeSH) terms. We restricted our search to English-language publications alone. Appendix A lists the search strategy used to search PubMed. In addition to searching for relevant articles, we also reviewed the references listed in the papers to avoid the risk of overlooking related studies that may satisfy the inclusion criteria, find all pertinent studies, and minimize publication bias. The search was based on the Preferred Reporting Items for Systematic Reviews and Meta-Analysis (PRISMA) Statement for Systematic Review and Meta-analysis [29] and the PRISMA extension of the NMA [30]. The literature search was conducted independently by two researchers (K.A. and Y.K.). Any disagreements that arose were resolved by discussions with a third author (T.Y.). The inclusion and exclusion criteria for the studies searched were collated using the Patients, Intervention, and Comparison, Outcome, and Study (PICOS) approach to address methodological or conceptual heterogeneity across studies and to secure the firmness of the NMA.

*2.2. Quality Evaluation*

We evaluated the qualities of the RCTs included in the NMA using the risk of bias tool 2 (RoB2) recommended by the Cochrane Collaboration [31]. We assessed the following parameters as being low risk, having some concerns, or high risk: (1) bias arising from the randomization process, (2) bias due to deviations from intended interventions, (3) bias due to missing outcome data, (4) bias in the outcome measurement, and (5) bias in the selection of the reported result.

*2.3. Inclusion and Exclusion Criteria (Predefined PICOS)*

2.3.1. Patients

The purpose of this study was to compare the efficacy and safety of ICIs+EP (Pem+EP, Dur+EP, or Atz+EP) and IP in adult patients with previously untreated ES-SCLC and to compare the efficacy and safety of ICI-containing regimens (Pem+EP, Dur+EP, or Atz+EP) with each other. We established inclusion criteria to accurately reflect the objectives of this study and included adult patients aged ≥18 years that were previously untreated, had histologically or cytologically confirmed ES-SCLC, and a performance status (PS) of 0–2 (on a 5-point scale, with higher values indicating greater disability).

2.3.2. Interventions/Comparisons

Patients received at least one of the following treatments: (1) Pem+EP, (2) Dur+EP, (3) Atz+EP, (4) AP, (5) IP, or (6) EP. We included PD-1/PD-L1 in our analysis, and we excluded regimens containing ICIs other than PD-1/PD-L1 from this analysis. The above regimens were adopted for ES-SCLC based on recommended dosages or those reported in phase III trials. EP was the commonly selected comparator as it is a standard therapeutic that was used before the emergence of molecular cancer therapy or immunotherapy for previously untreated ES-SCLC [3].

Network meta-analysis was performed not only for the four treatment groups, the ICIs+EP group (combined populations of Pem+EP, Dur+EP, and Atz+EP groups), AP group, IP group, and EP group, but also for the six groups of Pem+EP, Dur+EP, Atz+EP, AP, IP, and EP to compare between each pair of treatment groups.

2.3.3. Outcomes

The primary and secondary efficacy endpoints were OS and PFS, respectively, which were expressed in terms of hazard ratio (HR) and 95% credible interval (CrI). The primary safety endpoint was the incidence of ≥grade 3 adverse events (G3-AEs), which was expressed as risk ratio (RR) and 95% CrI. The secondary safety endpoints were the incidence of ≥grade 3 neutropenia (G3-NP), ≥grade 3 anemia (G3-AN), ≥grade 3 thrombocytopenia (G3-TP), and ≥grade 3 diarrhea (G3-diarrhea), which were expressed as RR and 95% CrI. To rank the efficacy and safety of each treatment, the surface under the cumulative ranking curve (SUCRA) values were calculated for each outcome. These predefined endpoints were only analyzed if data were available from the studies included.

2.3.4. Study Design

For inclusion in this NMA, randomized, parallel design trial phase III studies were eligible. The exclusion criteria included trials on children, observational studies, case reports, and non-RCTs. A parallel design trial is defined as a type of clinical research in which for two separate predefined intervention groups (intervention A group and intervention B group), one group is given only intervention A and the other group is given only intervention B.

*2.4. Statistical NMA Method*

We conducted Bayesian NMA per a robustly established methodology developed by the National Institute for Health and Care [21–23,32,33]. There are two main statistical methodologies for NMA: the frequentist approach and the Bayesian approach. We

adopted the standard Bayesian model described by Dias et al. [34–36], which presupposes inconsistency and heterogeneity among the included studies. We estimated the posterior distribution for effect size by conducting Gibbs sampling based on the Markov Chain Monte Carlo method, adopting a noninformative prior distribution. We set the number of iterations to 50,000 and considered the first 10,000 as a burn-in sample to eliminate the effect of the initial value. Effect sizes were expressed as HR and RR with their 95% CrIs; when the 95% CrI did not include 1, the difference in the effect size between the treatment groups was considered statistically significant. We calculated the SUCRA values to estimate the ranking of efficacy and safety outcomes; these values ranged from 0% to 100%, with higher SUCRA values indicating a more favorable treatment outcome [37]. We used the Brooks–Gelman–Rubin (BGR) diagnostic method to perform a convergent diagnosis of all comparisons [38,39]. To confirm the convergence of the model, both visual diagnosis and BGR diagnostics were conducted. The analysis was performed using OpenBUGS 1.4.0 (MRC Biostatistics Unit, Cambridge Public Health Research Institute, Cambridge, UK), and STATA (ver. 14, StataCorp, College Station, TX, USA) was utilized for the graphical presentation of the results.

### 2.5. Sensitivity Analysis

When conceptual heterogeneity was noted among the included studies, a sensitivity analysis was conducted by excluding the studies that showed heterogeneity. This allowed us to evaluate whether the inclusion/exclusion of studies with conceptual heterogeneity affected the final conclusions.

### 2.6. Assessment of Heterogeneity and Inconsistency

We statistically assessed heterogeneity and inconsistency of included studies to ensure the robustness of this analysis. Heterogeneity was assessed by using $I^2$ statistics among studies with direct comparisons [35]. Heterogeneity was judged to be low when $I^2$ was <40%, moderate when $I^2$ was ≥40% and <60%, substantial when $I^2$ was ≥60% and <75%, and considerable when $I^2$ was >75% [31]. Additionally, global inconsistency in the overall network model was assessed by using the statistical global inconsistency test [16,18,22,34]. This test was used to test for significance, and a value of $p < 0.05$ was considered to indicate the presence of significant inconsistency [16,18,22]. For statistical analysis of heterogeneity and inconsistency, we used the "metan" command and the "network" command of the STATA (ver. 14, StataCorp, College Station, TX, USA), respectively.

### 2.7. Ethical Aspects

Informed consent and approval by the institutional review board were waived owing to the nature of the systematic review conducted in the present study.

## 3. Results

### 3.1. Systematic Review

Among the 3903 articles identified through systematic literature review (734 from PubMed [25], 474 from Embase [26], and 909 from the Cochrane Central Register of Controlled Trials (CENTRAL) [27], and 1786 from SCOPUS [28]) that met the search criteria, 2790 articles were selected after removing duplicates.

Applying the PICOS design approach reduced the number of included studies in the present NMA to 10 articles (totaling 3879 patients), of which one study each compared Pem+EP and EP [6], Dur+EP and EP [7], Atz+EP and EP [8], AP and EP [40], and AP and IP [41]. The other five studies compared the effects of IP and EP administration [11,15,42–44]. Figure 1 illustrates the study selection process, Table S1 lists the key inclusion criteria, and Table S2 shows the detailed principal characteristics of the studies included. The data obtained from these studies were sufficient to perform an NMA using the predefined primary efficacy endpoint (OS), secondary efficacy endpoint (PFS), and secondary safety endpoints (G3-NP, G3-AN, G3-TP, G3-diarrhea) but not using the primary

safety endpoint (G3-AEs). Therefore, the G3-AEs were compared among the five treatment groups—Pem+EP, Dur+EP, Atz+EP, IP, and EP. In the analyses, the preferred model convergence was verified both visually and using the BGR method [38,39]. Maps of the NMA are shown in Figures 2 and 3.

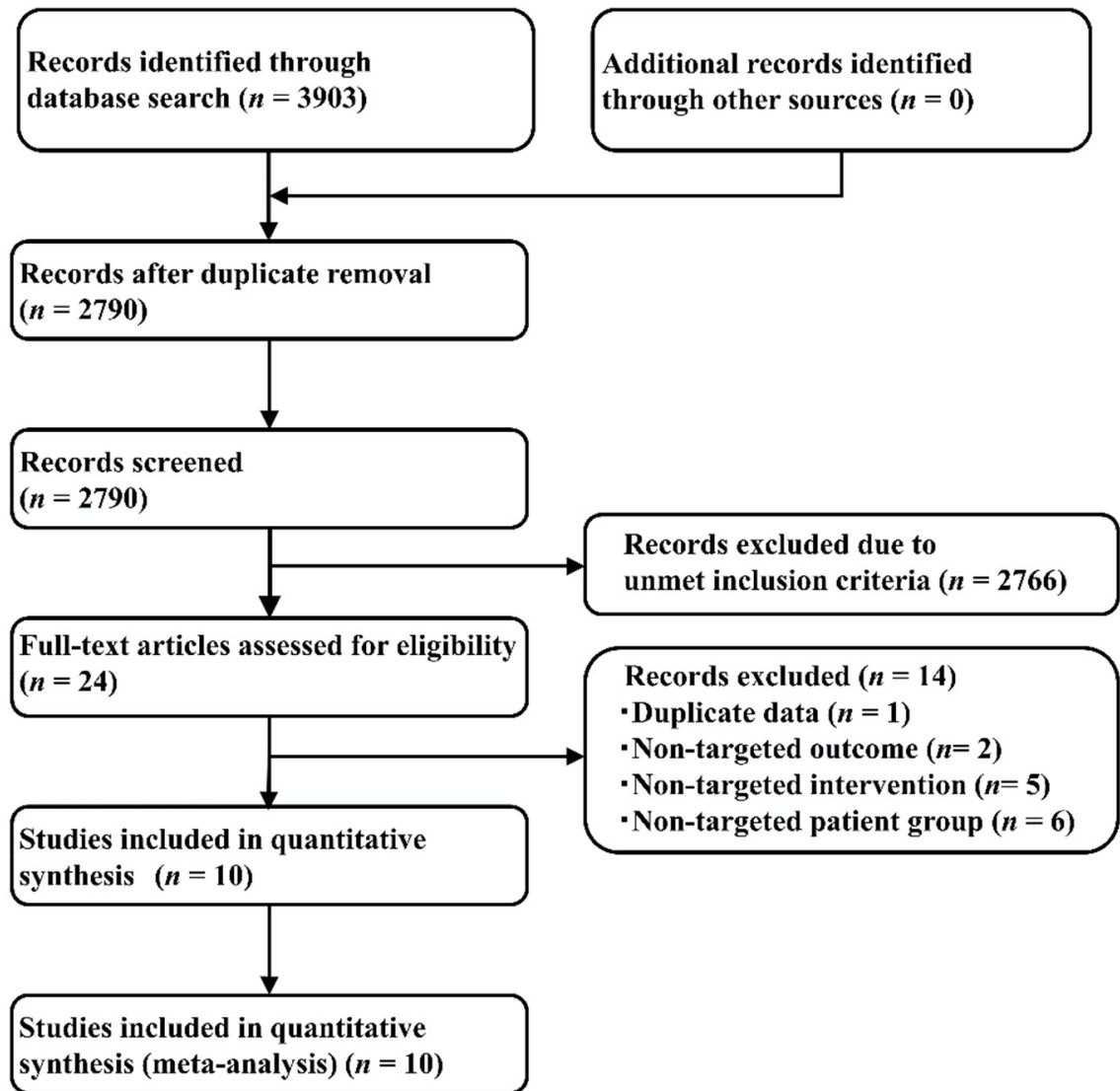

**Figure 1.** Study selection process.

### 3.2. Assessment of Risk of Bias and Heterogeneity/Inconsistency

We assessed the quality of the included studies using the Cochrane-recommended RoB2 [31]; no study was considered to have a high risk of bias, although four studies had some concerns for "bias due to deviations from intended interventions", as they were open-label studies (Figure S1).

In addition, we examined for potential heterogeneity and inconsistency in the NMA performed here. The heterogeneity of the direct comparison was calculated from the results of an integrated analysis of five studies [11,15,42–44] in a two-group comparison of IP vs. EP and from the results of an integrated analysis of three studies [6–8] in a two-group comparison of ICIs+EP vs. EP. The results were expressed as the $I^2$ statistic. The result was $I^2 = 33.2\%$ (IP vs. EP) and $I^2 = 0\%$ (ICIs+EP vs. EP), indicating that heterogeneity had little impact on the final conclusions (Figures S2 and S3).

Furthermore, global inconsistency in this NMA was evaluated using the statistical global inconsistency test. As a result, no significant inconsistency was detected ($p = 0.731$). Thus, we considered the heterogeneity or inconsistency to be unlikely to have influenced the final conclusions and that this NMA is valid.

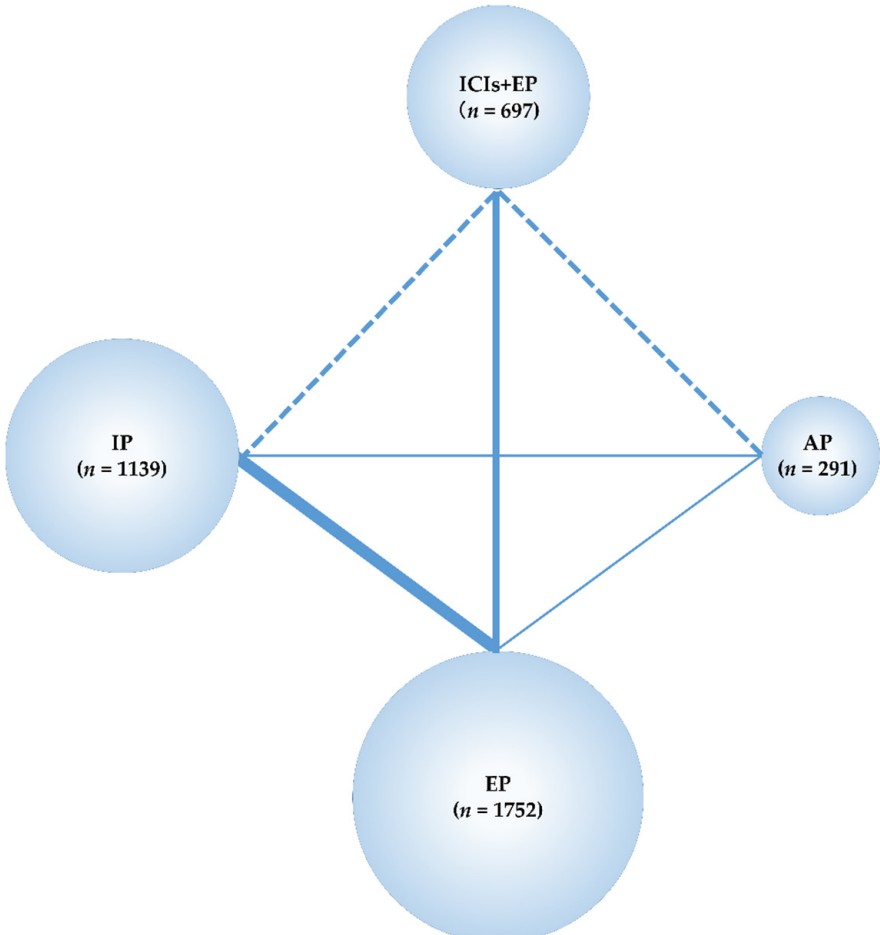

**Figure 2.** Network map of four treatment arms of immune checkpoint inhibitors plus platinum–etoposide (ICIs+EP) (combined treated group of pembrolizumab (Pem)+EP, durvalumab (Dur)+EP, and atezolizumab (Atz)+EP), amrubicin (AP), irinotecan (IP), and EP. The randomized controlled trials (RCTs) included in the network meta-analysis (NMA) are indicated by solid lines, and the width of the solid line corresponds to the number of studies included. The dashed line indicates the absence of head-to-head RCTs and that treatment comparisons may be attempted; n, number of patients included in each treatment group. ICIs+EP, immune check point inhibitors plus platinum–etoposide; Pem+EP, pembrolizumab plus platinum–etoposide; Dur+EP, durvalumab plus platinum–etoposide; Atz+EP, atezolizumab plus platinum–etoposide; AP, platinum–amrubicin; IP, platinum–irinotecan; EP, platinum–etoposide; RCT, randomized controlled trial; NMA, network meta-analysis.

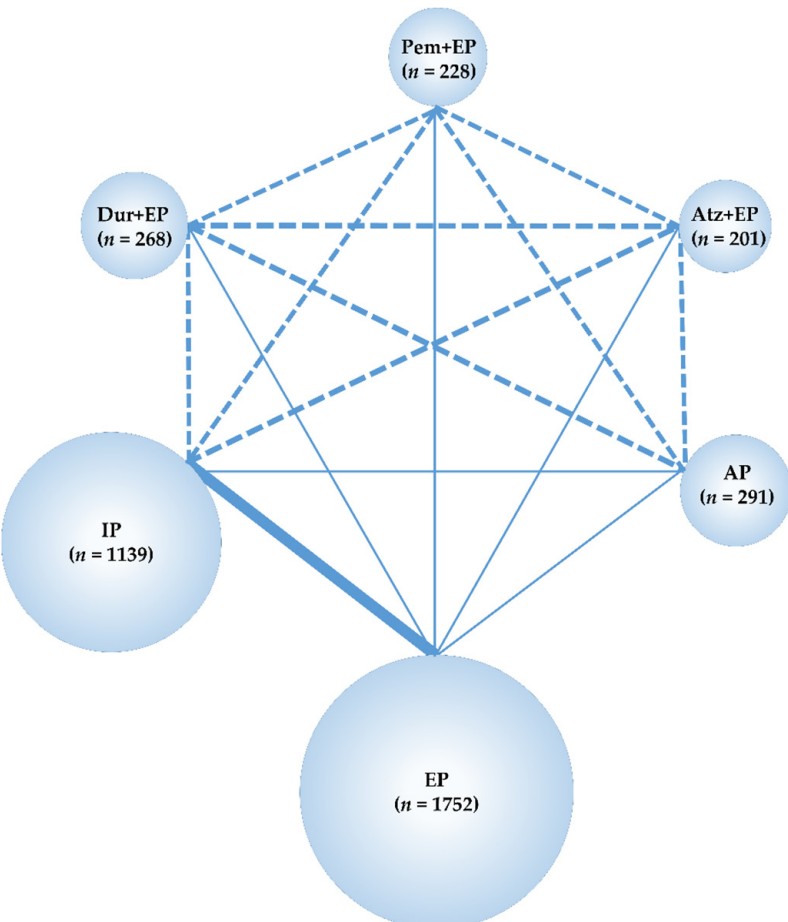

**Figure 3.** Network map of six treatment arms of three ICIs+EP regimens (Pem+EP, Dur+EP, and Atz+EP), AP, IP, and EP. The RCTs included in the NMA are indicated by solid lines, and the width of the solid line corresponds to the number of studies included. The dashed line indicates the absence of head-to-head RCTs and that treatment comparisons may be attempted; n, number of patients included in each treatment group. ICIs+EP, immune check point inhibitors plus platinum–etoposide; Pem+EP, pembrolizumab plus platinum–etoposide; Dur+EP, durvalumab plus platinum–etoposide; Atz+EP, atezolizumab plus platinum–etoposide; AP, platinum–amrubicin; IP, platinum–irinotecan; EP, platinum–etoposide; RCT, randomized controlled trial; NMA, network meta-analysis.

### 3.3. OS as the Primary Efficacy Endpoint

The primary efficacy endpoint of OS was compared between each pair of treatments across four treatment groups, including ICIs+EP (combined population of Atz+EP treated group, Dur+EP treated group, and Pem+EP treated group), AP, IP, and EP.

The results revealed no significant difference in OS between ICIs+EP group and IP group, with HR (95% CrI) of 0.896 (0.761–1.047), whereas OS was significantly better in ICIs+EP group than EP or AP group, with HRs (95% CrI) of 0.749 (0.655–0.852) and 0.772 (0.612–0.961), respectively (Figure 4).

To compare the efficacy in OS between each pair of three ICIs+EP regimens, an analysis of six treatment arms of three ICIs+EP regimens (Pem+EP, Dur+EP, Atz+EP), AP, IP, and EP was performed. The OS was significantly higher for patients treated with IP than for those treated with EP, with a HR (95% CrI) of 0.838 (0.765–0.916). The OS in the groups treated with Atz+EP, Dur+EP, or Pem+EP was significantly higher than that in groups treated with EP, with HRs (95% CrI) of 0.706 (0.539–0.908), 0.734 (0.588–0.906), and 0.805 (0.647–0.990), respectively. OS of groups treated with Atz+EP or Dur+EP was significantly higher than that of groups treated with AP, with HRs (95% CrI) of 0.728 (0.521–0.989) and 0.757 (0.565–0.996), respectively, whereas no significant differences in OS were ob-

served between any other pairs of treatments—i.e., the groups receiving AP and EP, AP and IP, Atz+EP and IP, Dur+EP and IP, Pem+EP and IP, Pem+EP and AP, Dur+EP and Atz+EP, Pem+EP and Atz+EP, and Pem+EP and Dur+EP (Figure S4).

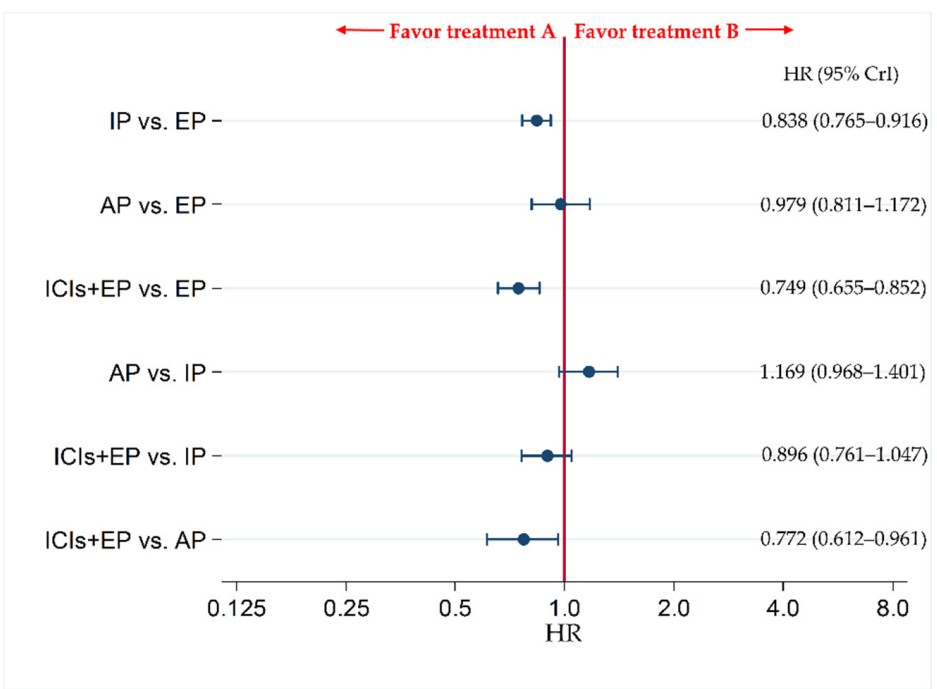

**Figure 4.** Comparative efficacy in terms of overall survival (OS) of each pair of treatments across four treatment groups, including ICIs+EP (combined population of Atz+EP treated group, Dur+EP treated group and Pem+EP treated group), AP, IP, and EP, for previously untreated extensive-stage small cell lung cancer (ES-SCLC). Comparisons are shown as treatment A vs. treatment B. Data are expressed as hazard ratio (HR) and 95% credible interval (CrI). ICIs+EP, immune checkpoint inhibitors plus platinum–etoposide; Atz+EP, atezolizumab plus platinum–etoposide; Dur+EP, durvalumab plus platinum–etoposide; Pem+EP, pembrolizumab plus platinum–etoposide; AP, platinum–amrubicin; IP, platinum–irinotecan; EP, platinum–etoposide; ES-SCLC, extensive-stage small cell lung cancer; CT, chemotherapy; HR, hazard ratio; CrI, credible interval.

Additionally, we ranked the efficacy and safety of the four treatment arms of ICIs+EP, AP, IP, and AP (Table S3), and of the six treatment arms of Pem+EP, Dur+EP, Atz+EP, AP, IP, and AP (Table S4) by assessing the surface under the cumulative ranking curve (SUCRA). Higher SUCRA values indicate better outcomes [37].

### 3.4. PFS as the Secondary Efficacy Endpoint

The secondary efficacy endpoint of PFS was compared between each pair of treatments across four treatment groups, including ICIs+EP (combined population of Atz+EP treated group, Dur+EP treated group, and Pem+EP treated group), AP, IP, and EP. The results revealed no significant difference in PFS between ICIs+EP group and IP group (HR: 0.889, 95% CrI: 0.766–1.025), whereas PFS was significantly better in ICIs+EP group than EP or AP group, with HRs (95% CrI) of 0.768 (0.683–0.860) and 0.702 (0.567–0.859), respectively (Figure 5).

To compare the efficacy in PFS between each pair of three ICIs+EP regimens, and analysis of six treatment arms of three ICIs+EP regimens (Pem+EP, Dur+EP, and Atz+EP), AP, IP, and EP was performed. The PFS with IP was significantly superior to EP, with an HR (95% CrI) of 0.866 (0.792–0.946). Additionally, PFS of the Atz+EP, Dur+EP, and Pem+EP groups was significantly improved compared with the EP group, with HRs (95% CrI) of 0.775 (0.618–0.959), 0.783 (0.649–0.938), and 0.754 (0.614–0.916), respectively. The PFS with

AP was significantly inferior to IP, with an HR (95% CrI) of 1.275 (1.078–1.499). In addition, PFS was also significantly higher in the Atz+EP, Dur+EP, and Pem+EP groups than in the AP group (HR (95% CrI): 0.708 (0.529–0.927), 0.716 (0.551–0.914), and 0.689 (0.524–0.889), respectively). No significant differences were observed in PFS between the AP and EP, Atz+EP and IP, Dur+EP and IP, Pem+EP and IP, Dur+EP and Atz+EP, Pem+EP and Atz+EP, and Pem+EP and Dur+EP groups (Figure S5).

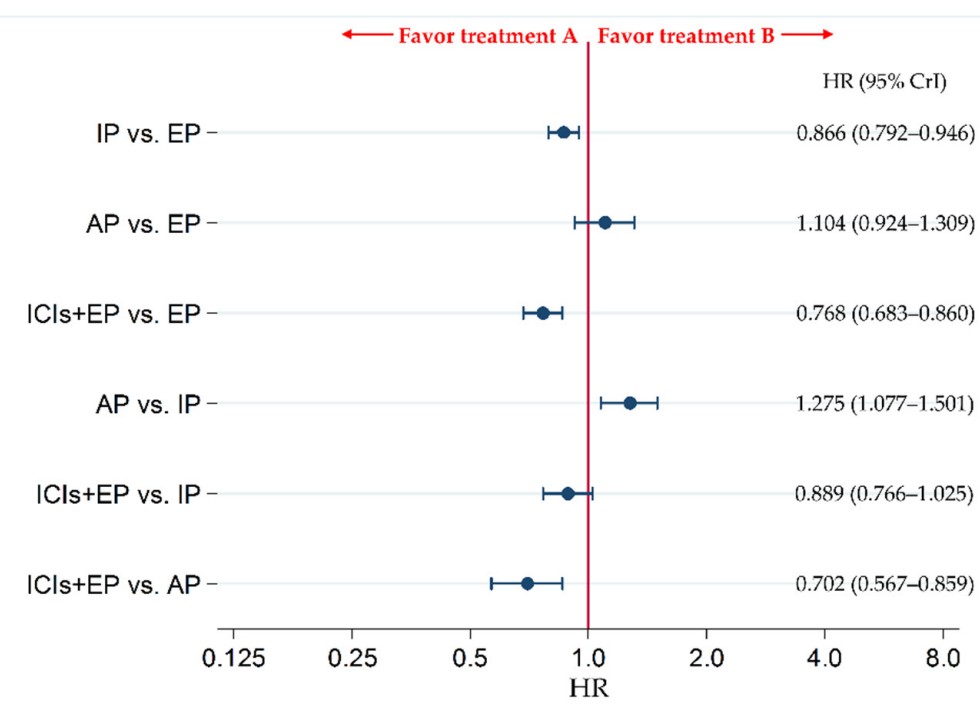

**Figure 5.** Comparative efficacy in terms of progression-free survival (PFS) of each pair of treatments across four treatment groups, including ICIs+EP (combined population of the treated groups Atz+EP, Dur+EP, and Pem+EP), EP, IP, and AP, for previously untreated ES-SCLC. Comparisons are shown as treatment A vs. treatment B. Data are expressed as HR and 95% CrI. ICIs+EP, immune checkpoint inhibitors plus platinum–etoposide; Atz+EP, atezolizumab plus platinum–etoposide; Dur+EP, durvalumab plus platinum–etoposide; Pem+EP, pembrolizumab plus platinum–etoposide; AP, platinum–amrubicin; IP, platinum–irinotecan; EP, platinum–etoposide; ES-SCLC, extensive-stage small cell lung cancer; CT, chemotherapy; HR, hazard ratio; CrI, credible interval.

### 3.5. The Incidence of G3-AEs as a Primary Safety Endpoint

The data for AP were insufficient for inclusion as a treatment group in the NMA for the primary safety endpoint of the incidence of G3-AEs. Therefore, the incidence of G3-AEs was compared between ICIs+EP (combined population of Pem+EP, Dur+EP, and Atz+EP treated group) and IP, and between ICIs+EP and EP. The results revealed no significant difference in incidence of G3-AEs between the ICIs+EP group and EP group, with an RR (95% CrI) of 0.983 (0.893–1.079), whereas the incidence of G3-AEs was significantly more frequent in the ICIs+EP group than the IP group, with an RR (95% CrI) of 1.262 (1.095–1.448) (Figure 6).

The NMA for G3-AEs was performed by including the five treatment regimens—Pem+EP, Dur+EP, Atz+EP, IP, and EP—and comparing each of the three ICIs+EP regimens with each other as the main objective. The incidence of G3-AEs in the IP group was significantly lower than in the EP group (risk ratio (RR) (95% CrI): 0.781 (0.704–0.866)). The incidence of G3-AEs in the Atz+EP group and Pem+EP group was significantly higher than in the IP group (RR (95% CrI): 1.301 (1.059–1.581) and 1.339 (1.113–1.598), respectively). No significant differences were observed in the incidence of G3-AEs between Atz+EP and



EP, Dur+EP and EP, Pem+EP and EP, Dur+EP and IP, Dur+EP and Atz+EP, Pem+EP and Atz+EP, and Pem+EP and Dur+EP groups (Figure S6).

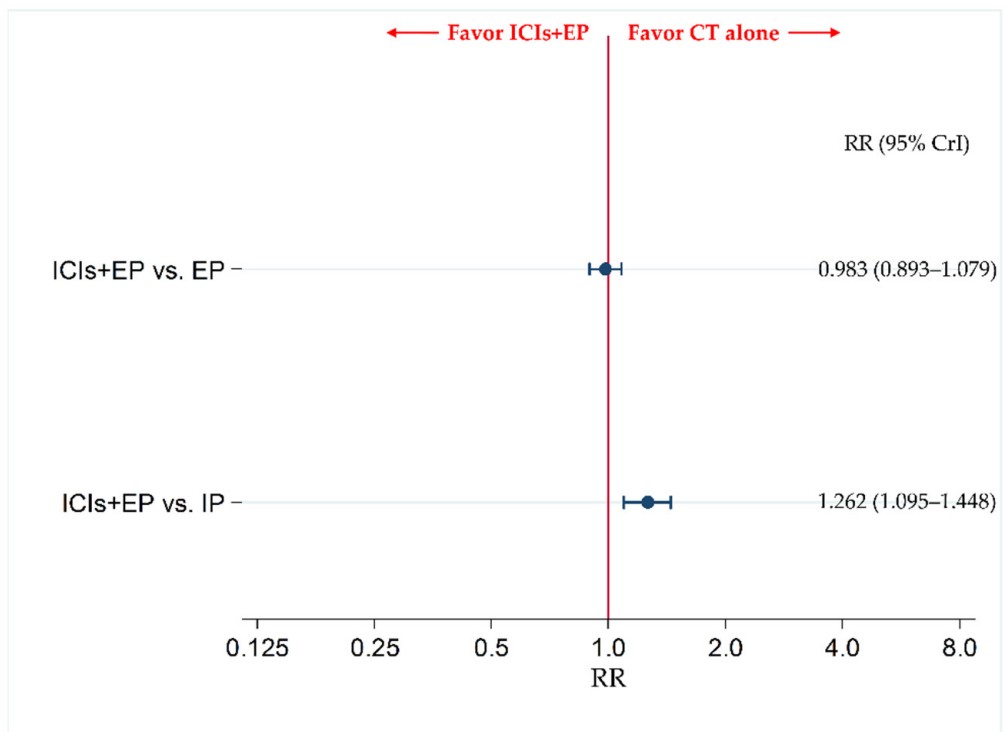

**Figure 6.** Comparative safety in terms of the incidence of ≥ grade 3 adverse events (G3-AEs) of each pair of treatments across three treatment groups, including ICIs+EP (combined population of Atz+EP treated group, Dur+EP treated group and Pem+EP treated group), EP, and IP, for previously untreated ES-SCLC. Comparisons are shown as ICIs+EP vs. CT (EP or IP). Data are expressed as risk ratio (RR) and 95% CrI. ICIs+EP, immune checkpoint inhibitors plus platinum–etoposide; Atz+EP, atezolizumab plus platinum–etoposide; Dur+EP, durvalumab plus platinum–etoposide; Pem+EP, pembrolizumab plus platinum–etoposide; EP, platinum–etoposide; IP, platinum–irinotecan; CT, chemotherapy; RR, risk ratio; CrI, credible interval.

*3.6. Secondary Safety Endpoint: The Incidence of G3-NP, G3-AN, and G3-TP*

The incidence of G3-NP, G3-AN, and G3-TP was compared between ICIs+EP (combined population of Pem+EP, Dur+EP, and Atz+EP treated group) and AP, between ICIs+EP and IP, and between ICIs+EP and EP.

The results revealed that there was no significant difference in G3-NP between the ICIs+EP group and EP or AP group, with RRs (95% CrI) of 0.900 (0.767–1.047) and 0.822 (0.666–1.004), respectively, whereas the incidence of G3-NP was significantly higher in ICIs+EP group than the IP group, with an RR (95% CrI) of 1.411 (1.181–1.670) (Figure 7a). The results also revealed that there was no significant difference in G3-AN between the ICIs+EP group and EP, IP, or AP group, with RRs (95% CrI) of 0.898 (0.666–1.182), 1.014 (0.687–1.442), and 0.689 (0.410–1.086), respectively (Figure 7b). There were no significant differences in the incidence of G3-TP between ICIs+EP and EP, with RRs (95% CrI) of 1.011 (0.710–1.392), whereas the incidence of G3-TP was significantly higher in the ICIs+EP group than that in the IP group, and was lower in the ICIs+EP group than that in the AP group, with RRs (95% CrI) of 2.205 (1.356–3.389) and 0.372 (0.178–0.688), respectively (Figure 7c).

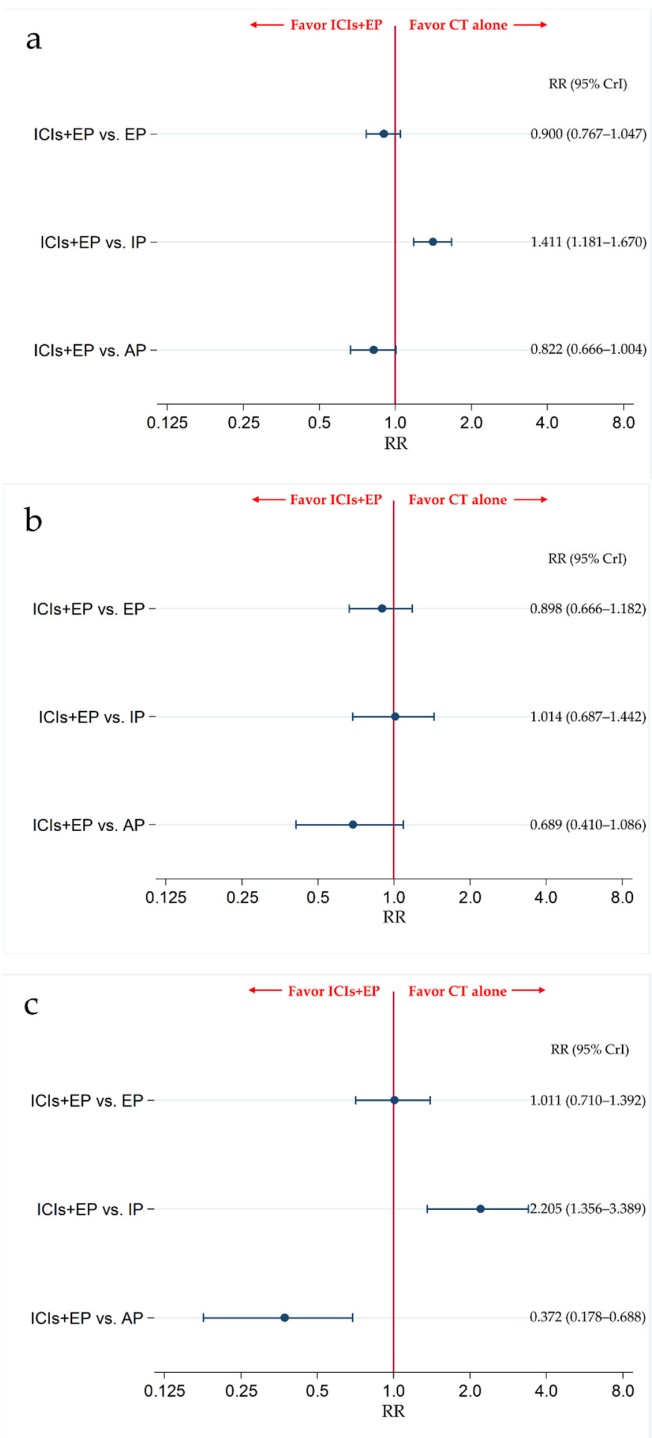

**Figure 7.** Comparative safety in terms of ≥grade 3 (**a**) neutropenia, (**b**) anemia, and (**c**) thrombocytopenia among the four treatment groups of ICIs+EP (combined group of Atz+EP treated group, Dur+EP treated group, and Pem treated group), EP, IP, and AP for previously untreated ES-SCLC. Comparisons are shown as ICIs+EP vs. CT. Data are expressed as RR and 95% CrI. ICIs+EP; immune checkpoint inhibitors plus platinum–etoposide; Atz+EP, atezolizumab plus platinum–etoposide; Dur+EP, durvalumab plus platinum–etoposide; Pem+EP, pembrolizumab plus platinum–etoposide; EP, platinum–etoposide; AP, platinum–amrubicin; IP, platinum–irinotecan; ES-SCLC, extensive-stage small cell lung cancer; CT, chemotherapy; RR, risk ratio; CrI, credible interval.

We also compared the safety in G3-NP, G3-AN, and G3-TP among six treatment arms of the Pem+EP group, Dur+EP group, Atz+EP group, AP group, IP group, and EP

group. The incidence of G3-NP in the IP group was significantly lower than in the EP group (RR: 0.639, 95% CrI: 0.591–0.688). G3-NP incidence was also significantly lower in the Dur+EP group than in the EP or AP group (RR (95% CrI): 0.717 (0.538–0.936), and 0.656 (0.476–0.882), respectively). The incidence of G3-NP in the AP, Atz+EP, and Pem+EP groups was significantly higher than in the IP group (RR (95% CrI): 1.722 (1.520–1.944), 1.514 (1.039–2.133), and 1.611 (1.265–2.023), respectively). Furthermore, the incidence of G3-NP was significantly higher in the Pem+EP group than in the Dur+EP group (RR: 1.461, 95% CrI: 1.004–2.048). However, no significant differences were observed in the incidence of G3-NP between AP and EP, Atz+EP and EP, Pem+EP and EP, Dur+EP and IP, Atz+EP and AP, Pem+EP and AP, Dur+EP and Atz+EP, and Pem+EP and Atz+EP (Figure S7a).

The incidence of G3-AN in the Dur+EP group was significantly lower than in the EP, AP, and Atz+EP groups (RR (95% CrI): 0.569 (0.329–0.917), 0.437 (0.217–0.793), 0.513 (0.231–0.989), respectively). The incidence of G3-AN in the AP group was significantly higher than in the IP group (RR: 1.513, 95% CrI: 1.061–2.099). No significant differences were observed in the incidence of G3-AN between the IP and EP, AP and EP, Atz+EP and EP, Pem+EP and EP, Atz+EP and IP, Dur+EP and IP, Pem+EP and IP, Atz+EP and AP, Pem+EP and AP, Pem+EP and Atz+EP, and Pem+EP and Dur+EP groups (Figure S7b).

The incidence of G3-TP in the IP group was significantly lower than in the EP group (RR: 0.470, 95% CrI: 0.341–0.632). The incidence of G3-TP in the AP group was significantly higher than in the EP group (RRs: 2.969, 95% CrI: 1.580–5.126). The incidence of G3-TP in the AP, Atz+EP, or Pem+EP groups was significantly higher than in the IP group (RRs (95% CrIs): 6.437 (3.286–11.46), 3.036 (1.399–5.788), and 2.618 (1.390–4.517), respectively). The incidence of G3-TP in the Dur+EP or Pem+EP group was significantly lower than in the AP group (RRs (95% CrIs): 0.229 (0.087–0.494)) and 0.442 (0.188–0.887), respectively). However, no significant differences were observed in the incidence of G3-TP between the Atz+EP and EP, Dur+EP and EP, Pem+EP and EP, Dur+EP and IP, Atz+EP and AP, Dur+EP and Atz+EP, Pem+EP and Atz+EP, and Pem+EP and Dur+EP groups (Figure S7c).

### 3.7. Secondary Safety Endpoint: The Incidence of G3-Diarrhea

The secondary efficacy endpoint of G3-diarrhea was compared between ICIs+EP (combined population of Pem+EP, Dur+EP, and Atz+EP treated group) and EP, between ICIs+EP and IP, and between ICIs+EP and AP. The results revealed no significant difference in G3-diarrhea between the ICIs+EP group and EP group and between the ICIs+EP and AP groups, with RRs (95% CrI) of 1.345 (0.525–2.840) and 0.842 (0.130–2.899), respectively, whereas G3-diarrhea was significantly lower in the ICIs+EP group than IP group, with an RR (95% CrI) of 0.156 (0.049–0.378) (Figure 8).

We also compared the safety in G3-diarrhea among six treatment arms of the Pem+EP group, Dur+EP group, Atz+EP group, AP group, IP group, and EP group. The incidence of G3-diarrhea in the IP group was significantly higher than that in the EP group (RR: 9.391, 95% CrI: 5.050–16.04). G3-diarrhea incidence was also significantly lower in the AP group, Dur+EP group, Pem+EP group than that in IP group (RR (95% CrI): 0.271 (0.064–0.773), 0.162 (0.021–0.611), and 0.137 (0.032–0.391), respectively). However, no significant differences were observed in the incidence of G3-diarrhea between the AP and EP, Atz+EP and EP, Dur+EP and EP, Pem+EP and EP, Atz+EP and IP, Atz+EP and AP, Dur+EP and AP, Pem+EP and AP, Dur+EP and Atz+EP, Pem+EP and Atz+EP, and Pem+EP and Dur+EP (Figure S8).

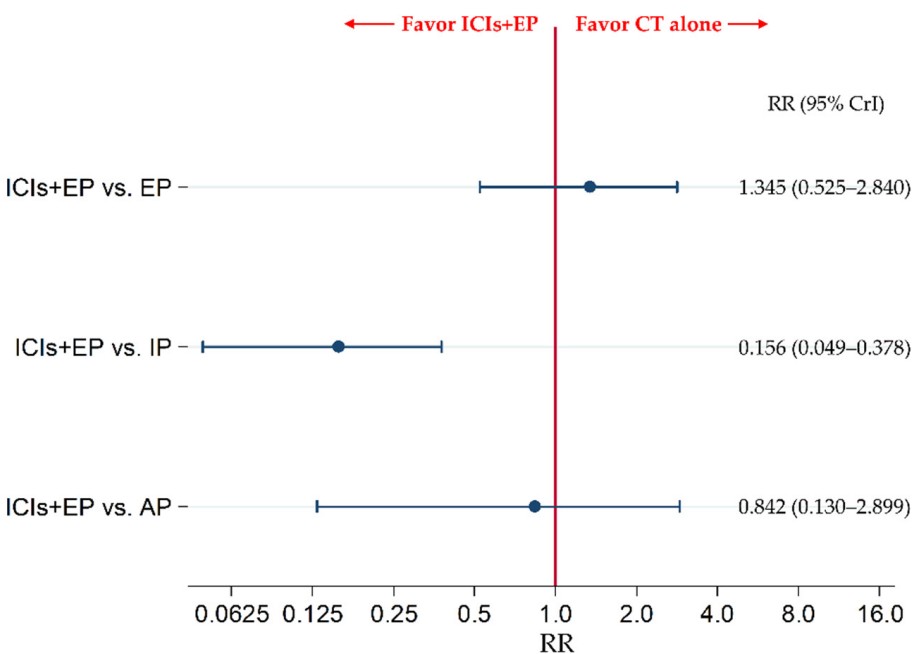

**Figure 8.** Comparative safety in terms of the incidence of ≥ grade 3 diarrhea between ICIs+EP (combined population of the treated groups of Atz+EP, Dur+EP, and Pem+EP) and EP, between ICIs+EP and IP, and between ICIs+EP and AP, for previously untreated ES-SCLC. Comparisons are shown as ICIs+EP vs. CT (EP, IP, or AP). Data are expressed as RR and 95% CrI. ICIs+EP, immune checkpoint inhibitors plus platinum–etoposide; Atz+EP, atezolizumab plus platinum–etoposide; Dur+EP, durvalumab plus platinum–etoposide; Pem+EP, pembrolizumab plus platinum–etoposide; EP, platinum–etoposide; IP, platinum–irinotecan; ES-SCLC, extensive-stage small cell lung cancer; CT, chemotherapy; RR, risk ratio; CrI, credible interval.

*3.8. Sensitivity Analysis*

Of the 10 trials included in this analysis, seven [6–8,11,40,41,43] limited their inclusion criteria to patients with PSs of 0 or 1, whereas the remaining three [15,42,44] also included patients with a PSs of 2 (Table S1). To assess the effect of this heterogeneity in the patient inclusion criteria of PS on the final conclusions, we conducted a sensitivity analysis by excluding the three trials [15,42,44] with the inclusion criteria that allowed the patients with PSs of 0 to 2 and we included the seven trials [6–8,11,40,41,43] with the inclusion criteria that only allowed patients with a PSs of 0 or 1. The sensitivity analysis assessed not only comparisons between ICIs+EP (combined population of Pem+EP, Dur+EP, and Atz+EP) and EP, ICIs+EP and IP, and ICIs+EP and AP, but also each pair of six treatment arms of Pem+EP, Dur+EP, Atz+EP, AP, IP, and EP. The results of the sensitivity analysis for primary endpoint, OS, showed that the significance of efficacy comparison between any pair of treatment groups remained unchanged. The SUCRA values and the ranks of each treatment group were similarly maintained. The detailed results of sensitivity analysis for comparison between ICIs+EP and EP, between ICIs+EP and IP, and between ICIs+EP and AP and for ranking assessment are shown in Tables S5 and S6, respectively. The detailed results of sensitivity analysis for comparison each pair of six treatment arms of Pem+EP, Dur+EP, Atz+EP, IP, and EP and for ranking assessment are shown in Tables S7 and S8, respectively.

Additionally, we considered that looking at geography as a variable was important. Of the 10 trials included in this analysis, six [6–8,11,42,43] were international cooperative studies or performed in Western countries, whereas the remaining four [15,40,41,44] had been performed in Asian countries. To assess the effect of this heterogeneity in geography on the final conclusions, we conducted a sensitivity analysis by excluding the four trials [15,40,41,44] performed in Asian countries, and included the six trials [6–8,11,42,43] which was international cooperative studies or performed in Western countries. None of

these six trials reported validation of AP as a treatment group. Therefore, the AP group could not be included in this sensitivity analysis. The sensitivity analysis assessed not only the results of comparison between ICIs+EP (combined population of Pem+EP and Dur+EP) and EP and between ICIs+EP and IP, but also the results of each pair of five treatment arms of Pem+EP, Dur+EP, Atz+EP, IP, and EP. The results of the sensitivity analysis for primary endpoint, OS, showed that the significance of efficacy comparison between any pair of treatment groups remained unchanged. The SUCRA values and the ranks of each treatment group were similarly maintained. The detailed results of sensitivity analysis for comparison between ICIs+EP and EP, between ICIs+EP and IP and for ranking assessment are shown in Tables S9 and S10, respectively. The detailed results of sensitivity analysis for comparison between each pair of five treatment arms of Pem+EP, Dur+EP, Atz+EP, IP, and EP and ranking assessment are shown in Tables S11 and S12, respectively.

Based on these results, we believe that the heterogeneity in PS and in geography for study inclusion between the studies included in this NMA did not affect the final conclusions.

## 4. Discussion

In this study, we compared the efficacy and safety profiles between ICIs+EP (combined population of three ICI-containing regimens) and IP for previously untreated ES-SCLC using Bayesian NMA; moreover, three ICI-containing treatment regimens (Pem+EP, Dur+EP, and Atz+EP) were compared with each other. In terms of OS and PFS, results showed no significant differences between ICIs+EP and IP. Furthermore, the incidence of G3-AEs was significantly higher in the ICIs+EP, the Pem+EP, and Atz+EP groups than in the IP group, with a more frequent incidence of G3-NP and G3-TP, whereas the incidence of G3-diarrhea in the ICIs+EP, Dur+EP, and Pem+EP groups was significantly lower than that in the IP group. The comparison of efficacy and safety among the three ICI-containing treatment regimens (Pem+EP, Dur+EP, and Atz+EP) showed no significant differences in primary efficacy or safety outcomes between any pair of treatment regimens. Regarding secondary safety outcomes, the incidence of G3-NP was significantly higher in the Pem+EP group than in the Dur+EP group, and the incidence of G3-AN was significantly lower in the Dur+EP group than in the Atz+EP group. These results may provide valuable information to clinicians regarding treatment strategies for previously untreated ES-SCLC.

Previous meta-analyses comparing ICIs plus chemotherapy with chemotherapy alone [45,46], and others comparing treatment regimens for previously untreated ES-SCLC [47–49] have reported that the ICIs+chemotherapy group had a better efficacy profile than the chemotherapy alone group. However, to date, no study has compared OS and PFS across six therapeutic regimens, including Pem+EP, Dur+EP, Atz+EP, AP, IP, and EP, for previously untreated ES-SCLC. Furthermore, no prior report assessed the details of efficacy and safety profiles of the ICIs+EP compared with IP. To the best of our knowledge, these results are the first to reveal that, although ICIs+EP had better OS and PFS than EP and AP, OS and PFS did not significantly differ between ICIs+EP and IP. Moreover, we found that the incidence of G3-AEs was significantly higher in ICIs+EP, in Pem+EP, or in Atz+EP than that in IP. In addition, the results revealed that the incidence of G3-NP or TP was significantly higher in ICIs+EP, in Atz+EP, or in Pem+EP than that in IP, whereas the incidence of G3-diarrhea was significantly lower in ICIs+EP, in Pem+EP, or in Dur+EP than that in IP. These new findings can help oncologists select more effective therapeutic strategies. Our results indicate that IP should be considered for patients for whom ICI is unavailable or unacceptable.

Although there were no significant differences in efficacy outcome between ICIs+EP and IP, ICIs+EP was superior to AP or EP, particularly Atz+EP or Dur+EP. The possible mechanisms underlying the combined effects of chemotherapy and immunotherapy are shown in Figure 9.

Chemotherapy induces apoptosis in cancer cells, but some chemotherapy-resistant cancer cells survive. Chemotherapy-induced apoptosis leads to the aggregation of activated T cells [50,51]. The anticancer activity of aggregated T cells is inhibited by PD-L1, which

is expressed by cancer cells; conversely, immunotherapy restores the anticancer activity of T cells by suppressing PD-L1 [50–52]. Owing to these mechanisms, chemotherapy and immunotherapy are more effective when used in combination than when used separately. In addition, the tumor mutation burden (TMB) is correlated with ICI sensitivity [53], and SCLC has been reported to exhibit higher TMB [3,4,9,10,54]. Although further investigation is warranted to determine whether TMB correlates with tumor susceptibility for ICI-containing regimens in ES-SCLC, a high TMB of SCLC may explain the increased efficacy of ICI-containing treatment regimens against ES-SCLC.

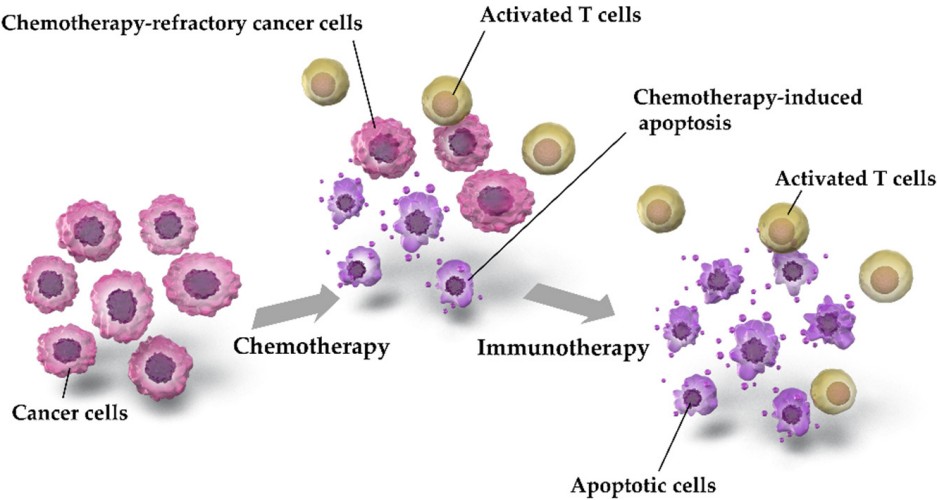

**Figure 9.** Possible mechanisms underlying the combination effects of chemotherapy and immunotherapy. Chemotherapy-induced apoptosis of cancer cells leads to the collection of activated T cells. Programmed Death-Ligand 1 (PD-L1)-mediated suppression of anticancer T cells by cancer cells is inhibited by treatment with ICIs, such as Pem, Dur, or Atz, which may restore T cell activation and induce cancer cell apoptosis. Accordingly, chemotherapy and immunotherapy are more effective when used as a combination therapy than when used separately.

In terms of ranking assessment, SUCRA values for OS showed that Atz+EP ranked the highest, followed by Dur+EP, Pem+EP, IP, AP, and EP. Regarding safety outcome, SUCRA values for G3-AE showed that IP ranked the safest, followed by Dur+EP, EP, Atz+EP, and Pem+EP (Table S4). We cannot exclusively conclude that all patients with previously untreated ES-SCLC should be treated with ICIs+EP. Rather, we believe that it is important to identify patients that may benefit most from each treatment strategy, considering both efficacy and safety.

Additionally, the results of the present analysis showed that G3-NP was significantly more frequent in Pem+EP than in Dur+EP, and G3-AN was significantly less frequent in Dur+EP than in Atz+EP. However, previous evidence in NSCLC suggests that there is no significant difference in the frequency of hematologic toxicity among ICIs [55–57]. Therefore, we do not believe that we can conclude from our study results that the frequency of hematologic toxicity in Dur+EP tends to be less than that in other ICIs+EP. Further studies to validate a more detailed profile of hematologic toxicity in Dur+EP are warranted.

The present study has several limitations. First, as this study was an NMA of RCTs performed separately, inconsistencies among the studies may have affected the results, although no statistically significant inconsistency was detected. Second, despite adaptation of the predefined PICOS approach, there were minor conceptual differences in the inclusion criteria of subjects for study inclusion in this NMA between the studies included in this analysis (i.e., differences in PS, in geography, and in platinum products, doses, and dosing schedules used). For example, seven of the ten trials had an inclusion criterion of patients with PSs of 0 or 1 [6–8,11,40,41,43]. Conversely, the inclusion criteria of patient PS in the remaining three trials additionally allowed for patients with PSs of 2 [15,42,44] (Table S1).

To address this heterogeneity, we conducted a sensitivity analysis by evaluating the effect of the inclusion/exclusion of these three trials on the final conclusions and found that they had an insignificant effect. Similarly, a sensitivity analysis was conducted using geography as a variable, but no change was observed in the final conclusion. However, the possibility of this heterogeneity effect might be unable to be completely ignored.

## 5. Conclusions

This study was conducted using Bayesian NMA statistical methods to evaluate and compare the efficacy and safety of ICIs+EP with that of IP as a first-line treatment for untreated patients with ES-SCLC. In addition, we compared the efficacy and safety of different ICIs+EPs with each other. The result revealed that no significant differences in OS and PFS were observed between ICIs+EP and IP and between each pair of three ICIs+EPs. The incidence of G3-AEs was significantly higher in ICIs+EP than IP, whereas there was no significant difference in the incidence of G3-AEs between each pair of the three ICIs+EPs. The G3-NP or G3-TP were significantly higher in ICIs+EP than that in IP, whereas the incidence of G3-diarrhea was significantly lower in ICIs+EP than IP.

These results may provide clinically relevant information regarding the efficacy and safety of these treatment regimens for previously untreated ES-SCLC. Considering that the present study is an NMA consisting of direct and indirect comparisons, verification via a direct head-to-head RCT is necessary to confirm our results. Furthermore, the characterization of patient profiles to identify the appropriate treatment regimen for each patient is an important topic for future clinical research.

**Supplementary Materials:** The following are available online at https://www.mdpi.com/1718-7729/28/2/106/s1: Figure S1: Risk of bias summary. Figure S2: Forest plot for five trials comparing IP and EP. Figure S3: Forest plot for three trials comparing ICIs+EP and EP. Figure S4: Comparative efficacy in terms of overall survival (OS) of each pair of treatments across six therapeutic regimens, including Pem+EP, Dur+EP, Atz+EP, AP, IP, and EP, for previously untreated ES-SCLC. Figure S5: Comparative efficacy for progression free survival (PFS) of each treatment pair across six therapeutic regimens, including Pem+EP, Dur+EP, Atz+EP, AP, IP, and EP, for previously untreated ES-SCLC. Figure S6: Comparative safety for $\geq$ grade 3 adverse events (G3-AEs) among the five therapeutic regimens—namely, Pem+EP, Dur+EP, Atz+EP, IP, and EP, for previously untreated ES-SCLC. Figure S7: Comparative safety in terms of $\geq$ grade 3 (a) neutropenia, (b) anemia, and (c) thrombocytopenia (c) among the six therapeutic regimens of Pem+EP, Dur+EP, Atz+EP, AP, IP, and EP for previously untreated ES-SCLC. Figure S8: Comparative safety for G3-diarrhea of each treatment pair across six therapeutic regimens, including Pem+EP, Dur+EP, Atz+EP, AP, IP, and EP, for previously untreated ES-SCLC. Table S1: Key inclusion criteria for included studies. Table S2: Characteristics of included studies. Table S3: Surface under the cumulative ranking curve (SUCRA) and (rank) of each of the four treatment regimens, including ICIs+EP, AP, IP, and EP for the efficacy and safety outcomes. Table S4: SUCRA (and rank) of each of the six treatment regimens, including Pem+EP, Dur+EP, Atz+EP, AP, IP, and EP for the efficacy and safety outcomes. Table S5: Sensitivity analysis for OS based on the performance statuses (PSs) of four treatment arms, including ICIs+EP, AP, EP, and IP. Table S6: Sensitivity analysis for ranking assessment for OS based on the performance statuses (PSs) of four treatment arms, including ICIs+EP, AP, EP, and IP. Table S7: Sensitivity analysis for OS based on the performance statuses (PSs) of six treatment arms, including Pem+EP, Dur+EP, Atz+EP, AP, EP, and IP. Table S8: Sensitivity analysis for ranking assessment for OS based on the performance status (PS) among six treatment arms, including Pem+EP, Dur+EP, Atz+EP, AP, EP, and IP. Table S9: Sensitivity analysis for OS based on the geography among three treatment arms, including ICIs+EP, EP, and IP. Table S10: Sensitivity analysis for ranking assessment for OS based on the geography among three treatment arms, including ICIs+EP, EP, and IP. Table S11: Sensitivity analysis for OS based on the geography of five treatment arms, including Pem+EP, Dur+EP, Atz+EP, EP, and IP. Table S12: Sensitivity analysis for ranking assessment for OS based on the geography among five treatment arms, including Pem+EP, Dur+EP, Atz+EP, EP, and IP.

**Author Contributions:** Conceptualization, K.A., R.M., Y.K., S.K., T.Y., A.T., T.O. (Tohru Ohmori), T.O. (Tsukasa Ohnishi) and H.S.; Data curation, K.A., R.M., Y.K., S.K., T.Y., A.T., T.O. (Tohru Ohmori), T.O.

(Tsukasa Ohnishi) and H.S.; Formal analysis, K.A., R.M., Y.K., S.K., T.Y., A.T., T.O. (Tohru Ohmori), T.O. (Tsukasa Ohnishi) and H.S.; Funding acquisition, K.A. and H.S.; Investigation, K.A., R.M., Y.K., S.K., T.Y., A.T., T.O. (Tohru Ohmori), T.O. (Tsukasa Ohnishi) and H.S.; Methodology, K.A., R.M., Y.K., S.K., T.Y., A.T., T.O. (Tohru Ohmori), T.O. (Tsukasa Ohnishi) and H.S.; Project administration, K.A., Y.K., S.K., T.Y., A.T., T.O. (Tohru Ohmori), T.O. (Tsukasa Ohnishi) and H.S.; Resources, K.A., R.M., Y.K., S.K. and T.Y.; Software, K.A.; Supervision, K.A., T.Y., A.T., T.O. (Tohru Ohmori), T.O. (Tsukasa Ohnishi) and H.S.; Validation, K.A., R.M., Y.K., S.K., T.Y., A.T., T.O. (Tohru Ohmori), T.O. (Tsukasa Ohnishi) and H.S.; Visualization, K.A.; Writing—original draft, K.A.; Writing—review and editing, K.A., Y.K., S.K., T.Y., A.T., T.O. (Tohru Ohmori), T.O. (Tsukasa Ohnishi) and H.S. All authors have read and agreed to the published version of the manuscript.

**Funding:** This research received no external funding.

**Acknowledgments:** We wish to express our gratitude to Hisashi Noma at the Institute of Statistical Mathematics and Toshiro Tango at the Medical Statistics Research Center for providing statistical support, and to Takashi Tsujino at the Science Graphics Inc. for providing graphic presentation support. We also thank the members of Division of Respiratory Medicine and Allergology, Department of Medicine, Showa University School of Medicine and the members of the Showa University Research Administration Center for their insightful comments and suggestions.

**Conflicts of Interest:** The authors declare no conflict of interest.

### Appendix A. Search Strategies in PubMed (Searched on 29 December 2020)

(("durvalumab"[ALL] OR "durvalumab"[Supplementary Concept] OR "imfinzi"[ALL] OR "MEDI-4736"[ALL] OR "atezolizumab"[ALL] OR "atezolizumab"[Supplementary Concept] OR "tecentriq"[ALL] OR "MPDL3280A"[ALL] OR "pembrolizumab"[ALL] OR "pembrolizumab"[Supplementary Concept] OR "Keytruda"[ALL] OR "MK-3475"[ALL] OR "immune checkpoint inhibitor*"[ALL] OR "anti-PD-1"[ALL] OR "anti-PD1"[ALL] OR "anti-PD 1"[ALL] OR "pd1 inhibitor*"[ALL] OR "pd 1 inhibitor*"[ALL] OR "pd 1 inhibitor nivolumab"[ALL] OR "programmed cell death 1 receptor/antagonists and inhibitors"[MeSH Terms] OR "programmed cell death 1 receptor antagonists and inhibitors"[ALL] OR "programmed cell death 1 receptor antagonist*"[ALL] OR "programmed cell death 1 receptor inhibitor*"[ALL]) AND (("cisplatin"[ALL] OR "cisplatin"[Supplementary Concept] OR "CDDP"[ALL] OR "carboplatin"[ALL] OR "carboplatin"[Supplementary Concept] OR "CBDCA"[ALL] OR "Platinum"[ALL]) AND ("etoposide"[ALL] OR "etoposide"[Supplementary Concept] OR "VP-16"[ALL] OR "Lastet"[ALL] OR "Vepesid"[ALL]))) OR (("etoposide"[ALL] OR "etoposide"[Supplementary Concept] OR "VP-16"[ALL] OR "Lastet"[ALL] OR "Vepesid"[ALL] OR "Irinotecan"[ALL] OR "Irinotecan"[Supplementary Concept] OR "CPT-11"[ALL] OR "Campto"[ALL] OR "Amrubicin"[ALL] OR "Amrubicin"[Supplementary Concept] OR "Calsed"[ALL]) AND ("cisplatin"[ALL] OR "cisplatin"[Supplementary Concept] OR "CDDP"[ALL] OR "carboplatin"[ALL] OR "carboplatin"[Supplementary Concept] OR "CBDCA"[ALL] OR "Platinum"[ALL])) AND (("Small Cell Lung Cancer"[Title/Abstract] OR "Small Cell Lung Carcinoma" [Title/Abstract] OR "Small-Cell Lung Carcinoma"[Title/Abstract] OR "Small-Cell Lung Carcinoma"[Title/Abstract] OR "SCLC"[Title/Abstract]) AND ("Extensive-stage"[Title/Abstract] OR "Extensive stage"[Title/Abstract] OR "Extensive-disease"[Title/Abstract] OR "Extensive disease"[Title/Abstract] OR "advanced"[Title/Abstract] OR "extensive"[Title/Abstract])) AND (Randomized Controlled trial[Title/Abstract] OR Controlled clinical trial[Title/Abstract] OR Randomized[Title/Abstract] OR Placebo[Title/Abstract] OR Randomly[Title/Abstract] OR Trial[Title/Abstract] OR Drug Therapy[Title/Abstract] OR Groups[Title/Abstract]).

A total of 734 results were obtained.

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
