# Peer review of "Comparative Efficacy and Safety of Immunotherapeutic Regimens with PD-1/PD-L1 Inhibitors for Previously Untreated Extensive-Stage Small Cell Lung Cancer: A Systematic Review and Network Meta-Analysis"

_curroncol, doi:10.3390/curroncol28020106_

Round 1
Reviewer 1 Report
The abstract could be improved. The author started the abstract without providing a short background on the subject.
Methods:
Did you choose a certain year period for the searches? What was the exclusion criteria?
Results:
Please double check if all your figures and tables have same letter styles. Consider putting some of the results data as supplementary (table or figure), because it seems a bit extensive. Figure 7 is nice, it can remain in the main text.
Check if you provided all the descriptions for the abbreviations used, since a lot were used in the text.
Overall: Good work, just need minor changes.
Author Response
Dear Reviewer 1
Thank you for giving us the opportunity to submit to Current Oncology a revised draft of our manuscript, “Comparative Efficacy and Safety of Immunotherapeutic Regimens with PD-1/PD-L1 Inhibitors for Previously Untreated Extensive-Stage Small-Cell Lung Cancer: A Systematic Review and Network Meta-Analysis” (Manuscript ID: curroncol-1087233). We sincerely appreciate your time and effort dedicated to providing valuable feedback on our manuscript. We are grateful to all three reviewers for their insightful comments, and we have incorporated changes to reflect most of their suggestions. These changes are highlighted in yellow below and within the manuscript.
Here is a point-by-point response to the reviewers’ comments and concerns.
Comment1: The abstract could be improved. The author started the abstract without providing a short background on the subject.
Response1: We agree. Thank you for your valuable suggestions. Following the reviewer's suggestion, we have added the following description of the background to the abstract section: “Improving therapeutic strategies for extensive-stage small-cell lung cancer (ES-SCLC) remains a challenge.”(L24–L25). We have also adjusted the entire abstract accordingly.  
Methods:
Comment2: Did you choose a certain year period for the searches?
Response2: Thank you for making this very important point. No restrictions were placed on the publication date in the electronic database used for the literature search, other than the publication date had to be after January 1, 1946. We have included the following statements in the relevant part of method section: “No restrictions were placed on the publication date in the electronic database used for the literature search, other than the publication date had to be after January 1, 1946.” (L111–L113).
Comment3: What was the exclusion criteria?
Response3: Thank you for raising this important point. The exclusion criteria included trials on children, observational studies, case reports, and non-randomized controlled trials. We have added this statement to the relevant part of the Methods section. (L198-L199).
Results:
Comment4: Please double check if all your figures and tables have same letter styles.
Response4: We agree. Thank you for addressing this very important point. We double-checked that all the figures and tables of the present manuscript have same letter styles. (The letter style of some figures was not able to be adjusted because the output of the statistical software is fixed to a particular letter style thereby not adjustable.)
Comment5: Consider putting some of the results data as supplementary (table or figure), because it seems a bit extensive.
Response5: We agree. Thank you for your valuable suggestion. The tables previously named Tables 1, 2, and 3 have been removed from the main manuscript and renumbered as Tables S1, S2, and S4, respectively, and presented in the Supplementary information. Similarly, the figures previously named Figures 3, 4, 5, and 6 were removed from the main manuscript and renumbered as Figures S2, S3, S4, and S5, respectively, and presented in the Supplementary information.
Comment6: Figure 7 is nice, it can remain in the main text.
Response6: Figure previously labeled as Figure 7 has been renumbered as Figure 9 and remains in the main text of the manuscript.
Comment7: Check if you provided all the descriptions for the abbreviations used, since a lot were used in the text.
Response7: Thank you for your valuable advice. We have ensured that we had provided the descriptions of all abbreviations used.
Overall: Good work, just need minor changes.
We are confident that our revised manuscript will be suitable for publication in Current Oncology and look forward to receiving your editorial decision.
Thank you for your consideration.
Sincerely,
Koichi Ando
Department of Medicine, Division of Respiratory Medicine and Allergology, Showa University School of Medicine
1-5-8 Hatanodai, Shinagawa-ku, Tokyo, 142-8666, Japan
Tel: +81-3-3784-8532
Fax: +81-3-3784-8742
Email: [email protected]

Reviewer 2 Report
Please see file.

Author Response
Dear Reviewer 2
Thank you for giving us the opportunity to submit to Current Oncology a revised draft of our manuscript, “Comparative Efficacy and Safety of Immunotherapeutic Regimens with PD-1/PD-L1 Inhibitors for Previously Untreated Extensive-Stage Small-Cell Lung Cancer: A Systematic Review and Network Meta-Analysis” (Manuscript ID: curroncol-1087233). We sincerely appreciate your time and effort dedicated to providing valuable feedback on our manuscript. We are grateful to all three reviewers for their insightful comments, and we have incorporated changes to reflect most of their suggestions. These changes are highlighted in yellow below and within the manuscript.
Here is a point-by-point response to the reviewers’ comments and concerns.
The authors performed a systematic with the goal of comparing efficacy and safety of ICI+EP to IP chemotherapy. This is an important question to examine considering ICIs may not be easily accessible in many parts of the world and I congratulate the authors on their work. However, there are some methodological points that need to be clarified and the presentation of results need to be simplified.
Comment1: Tables 1 and 2 can likely be combined. It would be helpful if the table can clearly state: 1) phase of RCT, 2) primary endpoint each trial
Response1: Thank you for your valuable suggestions. We agree with the reviewer's comments. However, it was difficult to combine the Tables previously represented by Table 1 and Table 2, partly because the page orientation for these tables had to be vertical. Instead, we removed both of these tables from the main manuscript, added the phase and primary endpoint information, and presented the table data in the Supplementary information as Table S1 and Table S2.
Comment2: Bias assessment should be presented much earlier in the results, rather than at the end.
Response2: We thank you for your valuable feedback. We agree with the reviewer's comments. In accordance with the reviewer's comment, we presented bias assessment much earlier in the results, rather than at the end (L322–L328, Figure S1).
Comment3: Heterogeneity was briefly discussed at the end and a sensitivity analysis was performed limiting patients to ECOG0-1. It is unclear from the methods section whether the sensitivity analysis was preplanned. In addition, there are other patient characteristics that contribute to heterogeneity. A more systematic approach in assessing heterogeneity of studies included or statistical tests of heterogeneity needs to be presented, especially since there are multiple comparisons.
Response3: We sincerely appreciate your comments. We agree with the reviewer's comments. It was assumed in the planning stage that sensitivity analysis would be conducted, and the necessary descriptions have been included in the appropriate places in the Methods section (L234–L240). In addition, we considered geography as a patient characteristic that affects heterogeneity, in addition to performance status. We performed a sensitivity analysis, excluding studies conducted in the Asian region. We have included relevant statements in the relevant part of the Results section (L603–L628). Furthermore, we conducted statistical tests of heterogeneity and inconsistency to confirm the validity of the present NMA. As a result, the heterogeneity did not affect the final conclusions and no significant inconsistency was detected. Therefore, we considered this NMA to be valid (L633–L646).
Comment4: The results section is extremely difficult to follow: there are 15 comparisons for each endpoint of OS, PFS, AE and each AE. Please show a summary of grouped results whenever possible - the primary objective to investigate ICI+EP vs IP. It is much easier to comprehend if you group all ICIs together, especially since you showed that the ICIs were not different.
Response4: Thank you for your very important suggestion. We agree with the reviewer's comments. In response, we combined the Pem+EP treated group, Dur+EP treated group, and Atz+EP treated group into the ICIs+EP treated group. We then compared these groups for each outcome (OS, PFS, G3-AEs, G3-NP, G3-AN, G3-TP, G3-diarrhea) between ICIs+EP and EP, ICIs+EP and IP, and ICIs+EP and AP, and presented the results in the main manuscript (in Figures 4-8). Accordingly, the figures that previously appeared as Figures 3, 4, 5, and 6a-c have been included only in the Supplement and are now numbered as Figures S2, 3, 4, and 5a-c, respectively.
Comment5: Are hematologic AEs from Durva-EP actually different from Pembro-EP? Or is this simply due to chance? Is this clinically relevant? The evidence on chemoIO in NSCLC would suggest that most of the hematologic AEs come from the chemotherapy and that there are no significant differences among ICIs.
Response5: Thank you for raising this important point. We agree with the reviewer's comments. Our results indeed showed that the frequency of G3-NP was significantly higher in Pem+EP than in Dur+EP, and the frequency of G3-AN was significantly lower in Dur+EP than in Atz+EP. However, as you point out, the evidence on ICI in NSCLC suggests there is no significant difference in hematologic toxicity among ICIs. Therefore, we believe that the comparison between each of the ICIs+Ep for the hematological toxicity in this study needs further validation. Thus, the results of the comparative analysis of G3-NP, G3-AN, and G3-TP among Pem+EP, Dur+EP, and Atz+EP could not be included in the final conclusions (in the Conclusion section), but were only discussed in the corresponding part of the Discussion section as follows: “Also, the results of the present analysis showed that G3-NP was significantly more frequent in Pem+EP than in Dur+EP, and G3-AN was significantly less frequent in Dur+EP than in Atz+EP. However, previous evidence in NSCLC suggests that there is no significant difference in the frequency of hematologic toxicity among ICIs [55-57]. Therefore, we do not believe that we can conclude from our study results that the frequency of hematologic toxicity in Dur+EP tends to be less than that in other ICIs+EP. Further studies to validate a more detailed profile of hematologic toxicity in Dur+EP are warranted.” (L728–L737).
Comment6: It is not necessary to separate all the hematologic AEs. But grade 3 diarrhea should be included as an AE of interest, given that it is higher with irinotecan.
Response6: Thank you very much for your very valuable suggestion. We agree with the reviewer's comments. In the included studies of the present analysis, the incidence of all the hematologic AEs was not reported as an outcome and could not be adopted as a primary or secondary endpoint in our analysis. Therefore, we compared the frequencies of G3-NP, G3-AN, and G3-TP between ICIs+EP (a combination of the groups Pem+EP, Dur+EP, and Atz+EP) and AP, IP, and EP, respectively. The results are shown in L459–L479
Also, we examined the frequency of grade 3 or higher diarrhea (G3-diarrhea) as a secondary endpoint in an additional analysis. Comparisons were made between ICIs+EP (combined populations of Pem+EP, Dur+EP, and Atz+EP) and AP, IP, and EP, respectively. In addition, comparisons in the incidence of G3-diarrhea were made between each pair of the 6 treatment groups (Pem+EP, Dur+EP, Atz+EP. AP, IP, EP) respectively. The results are shown in L538-L546, Figure 8, and Figure S6.
Comment7: Figure 7 seems a little out of place.
Response7: Thank you for making this important point. We also apologize for not fully explaining the intent of the figure previously shown as Figure 7 (now Figure 9 in the revised manuscript). In our results, ICIs+EP had significantly better OS and PFS compared to AP or EP, although not significantly different from IP. We thought it necessary to provide a molecular biological explanation for the superiority in OS and PFS of ICIs+EP over AP or EP; we therefore presented these data in the original Figure 7 (Figure 9 in the revised manuscript). We have added the necessary description in the text to explain this figure as follows: “Although there were no significant differences in efficacy outcome between ICIs+EP and IP, ICIs+EP is superior to AP or EP, particularly Atz+EP or Dur+EP. The possible mechanisms underlying the combined effects of chemotherapy and immunotherapy are shown in Figure 9.” (L690–L694).
Comment8: The results quite clearly showed superior efficacy of ICI+EP vs IP at the cost of increased toxicity; but very difficult to tease out because of multiple comparisons. It would be very helpful if you can show grouped comparisons of ICI+EP vs IP for OS, any grade 3 AE, grade 3 hematologic AE and grade 3 non-hematologic AE
Response8: Thank you sincerely for your very valuable suggestion. We agree with the reviewer's comment. In response, the Pem+EP treated group, Dur+EP treated group, and Atz-treated group have been merged into ICIs+EP. For each outcome (OS, PFS, G3-AEs, G3-NP, G3-AN, G3-TP, G3-diarrhea), ICIs+EP was compared with AP, IP, and EP, respectively. In the revised manuscript, the network map was presented in Figure 2, and the compared results were presented in Figures 4, 5, 6, 7, and 8.
The studies included in our analysis adopted any grade 3 AE as the outcome. However, grade 3 hematologic AE and grade 3 non-hematologic AE were not adopted as outcomes. Therefore, in our analysis, although any grade 3 AE was included as a primary endpoint (the results were shown in Figure 6), grade 3 hematologic AEs and non-hematologic AEs could not be included in the primary or secondary endpoint. We adopted G3-NP, G3-AN, and G3-TP as endpoints to validate hematologic AEs. In addition, we examined the frequency of G3-diarrhea as a grade 3 non-hematologic AE. The results are shown in Figures 7 and 8 in the revised manuscript.
We are confident that our revised manuscript will be suitable for publication in Current Oncology and look forward to receiving your editorial decision.
Thank you for your consideration.
Sincerely,
Koichi Ando
Department of Medicine, Division of Respiratory Medicine and Allergology, Showa University School of Medicine
1-5-8 Hatanodai, Shinagawa-ku, Tokyo, 142-8666, Japan
Tel: +81-3-3784-8532
Fax: +81-3-3784-8742
Email: [email protected]

Reviewer 3 Report
I would clarify that you are specifically talking about PD1/PDL1 inhibitors as opposed to all ICI
46-48 - The wording needs to be changed. Sounds like small cell contributes 15% of ALL cancer deaths which is incorrect.
61-63- In North America IP is not used as commonly as EP based on SWOG S0124 PMID19349543
161- Define G3-NP/AN/TP if not already done
168 - Parallel group is not a term I am familiar with. I understand what you are saying.
Can you make Table 2 vertical. Listing ES-SCLC is redundant
Looking at Figure 3 ICI+EP comparisons are simply cross trial comparisons. The network meta-analysis doesn't add anything to simply looking numerically at the various hazard ratios.
Is this actually a valid Network? All of the trials are being compared to EP. EP vs IP has 6 trials so a meta-analysis is appropriate. All other "nodes" are connected directly to EP. Atez+EP vs EP HR 0.706, IP vs EP HR .838. Looking at the fraction of these two HRs essentially predicts the "network result". 0.706/0.838=0.842 - Network result 0.845
I also find section 3.6 is not needed or should be reframed. SUCRA does not add clarity for clinicians
3.7 - was there any thought to looking at geography as a variable given that IP=EP in the SWOG study leading to lower uptake of IP in North America
Author Response
Dear Reviewer 3
Thank you for giving us the opportunity to submit to Current Oncology a revised draft of our manuscript, “Comparative Efficacy and Safety of Immunotherapeutic Regimens with PD-1/PD-L1 Inhibitors for Previously Untreated Extensive-Stage Small-Cell Lung Cancer: A Systematic Review and Network Meta-Analysis” (Manuscript ID: curroncol-1087233). We sincerely appreciate your time and effort dedicated to providing valuable feedback on our manuscript. We are grateful to all three reviewers for their insightful comments, and we have incorporated changes to reflect most of their suggestions. These changes are highlighted in yellow below and within the manuscript.
Here is a point-by-point response to the reviewers’ comments and concerns.
Comment1: I would clarify that you are specifically talking about PD1/PDL1 inhibitors as opposed to all ICI
Response1: Thank you very much for your valuable suggestion. We agree with the reviewer's comments. We have added the following statement to the Methods section to clarify that the ICIs analyzed were limited to PD-1/PD-L1 inhibitors as follows: "We included PD-1/PD-L1 in our analysis, and we excluded regimens containing ICIs other than PD-1/PD-L1 from this analysis.” (L165-L167).
Comment2: 46-48 - The wording needs to be changed. Sounds like small cell contributes 15% of ALL cancer deaths which is incorrect.
Response2: Thank you for noting this very important point. We agree with the reviewer's comments. We have corrected the relevant part as follows: “Lung cancer is responsible for most cancer-induced mortality worldwide, with small-cell lung cancer (SCLC) accounting for 15% of newly diagnosed lung cancer cases.” (L49-51).
Comment3: 61-63- In North America IP is not used as commonly as EP based on SWOG S0124 PMID19349543
Response3: Thank you for addressing a very important point. We agree with the reviewer's comments. As you pointed out, we have revised the relevant part as follows: “In North America, based on the results of the SWOG S0124 study, (platinum–irinotecan) IP is not as commonly used as EP, whereas in Japan, however, IP is currently one of the leading treatment options for previously untreated ES-SCLC based on the results of previous phase III studies and meta-analysis [12–15].” (L67-71).
Comment4: 161- Define G3-NP/AN/TP if not already done
Response4: Thank you for noting this very important point. We agree with the reviewer's comments. We have defined G3-NP/AN/TP as follows: “The secondary safety endpoints were the incidence of ≥ grade 3 neutropenia (G3-NP), ≥ grade 3 anemia (G3-AN), ≥ grade 3 thrombocytopenia (G3-TP), and ≥ grade 3 diarrhea (G3-diarrhea), which were expressed as RR and 95% CrI.”(L186-190).
Comment5: 168 - Parallel group is not a term I am familiar with. I understand what you are saying.
Response5: Thank you for bringing up an important point. We agree with the reviewer's comments. We have changed the term "parallel study" to the more accurate term " parallel design trial" (L197) and added the following explanation of the term: “A parallel design trial is defined as a type of clinical research in which for two separate pre-defined intervention groups (intervention A group and intervention B group) one group is given only intervention A and the other group is given only intervention B”.(L197-L203). 
Comment6: Can you make Table 2 vertical. Listing ES-SCLC is redundant.
Response6: Thank you for addressing a very important point. We agree with the reviewer's comments. We have changed the printing orientation of the Table that used to be Table 2 to vertical. We have excluded this table from the main manuscript, renumbered it as Table S2, and added new information about phase, primary endpoint, which is shown in the supplementary information.
Comment7: Looking at Figure 3 ICI+EP comparisons are simply cross trial comparisons. The network meta-analysis doesn't add anything to simply looking numerically at the various hazard ratios.
Response7: Thank you very much for focusing on this very important point. We agree with the reviewer's comments. As reviewer pointed out, simply calculating the effect size of the hazard ratio (or risk ratio) is insufficient to achieve the purpose of this study. The purpose of this Bayesian network meta-analysis was to compare the efficacy and safety of each treatment group. To make a comparison between the intervention groups---to assess for statistical significance---it is necessary to calculate a credible interval (95% credible interval in this study) according to a statistically valid methodology. Therefore, the network meta-analysis statistical methodology focuses on estimating statistically valid credible intervals as well as point estimates of the effect size. To show this focus more clearly, we have revised the relevant sections as follows: “when the 95% CrI did not include 1, the difference in the effect size between the treatment groups was considered statistically significant.” (L219-L221).
Comment8: Is this actually a valid Network? All of the trials are being compared to EP. EP vs IP has 6 trials so a meta-analysis is appropriate. All other "nodes" are connected directly to EP.
Response8: Thank you for your very valuable comments. One of the advantages of network meta-analysis is that effect size can be estimated using common comparators, even if there are no previously reported randomized trials that directly compared the two. For example, although no previous randomized controlled trials have directly compared Atz+EP and IP, it is possible to compare them using EP as a common comparator from trials that compared between Atz+EP and EP, and between IP and EP. Therefore, the network is considered valid with the use of appropriate analysis methodologies. Another advantage of network meta-analysis is that it may provide a better estimate of the effect of interventions by incorporating not only evidence from direct comparison but also evidence from indirect comparison. For example, the comparison between EP and IP incorporated not only evidence from direct comparisons between EP and IP, but also indirect evidence from the literature comparing between EP and AP, and between IP and AP. Thus, we intended to obtain a better estimate by combining evidence from direct comparisons with evidence from indirect comparisons through network meta-analysis in this study. In general, however, the robustness of a network meta-analysis assumes that no significant inconsistency between each study has been incorporated into the network. We statistically assessed whether any significant inconsistency was present among the included studies in this network meta-analysis. As a result, no significant inconsistency was detected in this network meta-analysis. In addition, we performed a meta-analysis of trials directly comparing EP and IP (5 trials) with OS as the outcome to examine heterogeneity. The statistical heterogeneity I2 statistic was used to express the results. The results are shown in Figure S7 in Supplementary. The I2 statistic was 33.2%, which indicates that heterogeneity did not affect the final conclusions. Therefore, we considered this NMA to be valid (L633-L646).
Comment9: Atez+EP vs EP HR 0.706, IP vs EP HR .838. Looking at the fraction of these two HRs essentially predicts the "network result". 0.706/0.838=0.842 - Network result 0.845
Response9: This study adopted the Bayesian approach as a statistical method of NMA; however, as you presented, the frequentist approach is also known as a statistical methodology for NMA, and both have been reported to show approximate effect sizes. We have described this choice of statistical methodology more clearly in the Methods section as follows: ”There are two main statistical methodologies for NMA: the frequentist approach and the Bayesian approach. We adopted the standard Bayesian model described by Dias et al. [34–36], which presupposes inconsistency and heterogeneity among the included studies.”(L209-213).
Comment10: I also find section 3.6 is not needed or should be reframed. SUCRA does not add clarity for clinicians
Response10: I would like to thank you for your very important remarks. We agree with the reviewer's comments. As you pointed out, SUCRA is not an absolute criterion for clinicians to choose a treatment regimen. As per your suggestion, we have deleted Section 3.6, and Table 3 is now only represented in the Supplementary information as Table 4S. As a result, the description of Table 3 in the text has been changed to Table S4. Furthermore, the results of SUCRA were not included in the basis for our final conclusions.
Comment11: 3.7 - was there any thought to looking at geography as a variable given that IP=EP in the SWOG study leading to lower uptake of IP in North America.
Response11: We sincerely thank you for your focus on a very important point. We agree with the reviewer's comments. As you pointed out, I think geography is a very important variable to consider. To examine the impact of geography on our final conclusions, we conducted an additional sensitivity analysis that excluded four studies conducted in Asia [15,40,41,44] and included six studies that had been international cooperative study or performed in Western countries [6–8,11,42,43]. Because the six included studies did not include the AP treatment group, AP could not be included in the analysis; therefore, we examined five treatment groups (Atz+EP, Dur+EP, Pem+EP, IP, EP). The findings showed that results of analysis for significance remained similar for all comparisons. The results of the ranking of each treatment group were also maintained, with SUCRA showing similar values for each treatment group. The results of this sensitivity analysis indicate that geography did not affect the final conclusions of the present study. We have included the results of this sensitivity analysis in the relevant part of Results section 3.8 Sensitivity analysis (L603-L628).
We are confident that our revised manuscript will be suitable for publication in Current Oncology and look forward to receiving your editorial decision.
Thank you for your consideration.
Sincerely,
Koichi Ando
Department of Medicine, Division of Respiratory Medicine and Allergology, Showa University School of Medicine
1-5-8 Hatanodai, Shinagawa-ku, Tokyo, 142-8666, Japan
Tel: +81-3-3784-8532
Fax: +81-3-3784-8742
Email: [email protected]

Round 2
Reviewer 2 Report
These revisions are excellent. The figures are much easier to follow. Minor suggestions:
1) Suggest adding back the chemotherapy comparisons (EP vs IP, AP vs IP, PE vs AP) to Figures 4 and 5. Since the journal audience is primarily North American, readers may be less familiar with the IP literature in SCLC. The results indicate that IP should be considered if ICI is not available or accessible.
2) Combine the assessments for bias/heterogeneity/inconsistency assessment into section 3.2
3) Assessment of heterogeneity - what about the EP+ICI vs EP studies?
Author Response
Dear Reviewer 2
We are very pleased to know that our manuscript titled "Comparative Efficacy and Safety of Immunotherapeutic Regimens with PD-1/PD-L1 Inhibitors for Previously Untreated Extensive-Stage Small-Cell Lung Cancer: A Systematic Review and Network Meta-Analysis” (Manuscript ID: curroncol-1087233) will be accepted for publication upon the completion of minor revisions. We sincerely appreciate the time and effort you have dedicated to provide valuable feedback on our manuscript. We are grateful to all three reviewers for their insightful comments, and we have incorporated changes that reflect most of their suggestions. These changes are highlighted in yellow below and within the manuscript.
These revisions are excellent. The figures are much easier to follow. Minor suggestions:
Comment1: Suggest adding back the chemotherapy comparisons (EP vs IP, AP vs IP, PE vs AP) to Figures 4 and 5. Since the journal audience is primarily North American, readers may be less familiar with the IP literature in SCLC. The results indicate that IP should be considered if ICI is not available or accessible.
Response1: Thank you for your very valuable suggestion. Following your comments, I have added the results of the comparisons for IP vs. EP, AP vs. EP, and AP vs. IP to Figures 4 and 5. We have also added the following statement to the relevant part of the discussion section: “Our results indicate that IP should be considered for patients for whom ICI is unavailable or unacceptable.” (L678–L680)
Comment2: Combine the assessments for bias/heterogeneity/inconsistency assessment into section 3.2
Response2: Thank you very much for highlighting this very useful point. Per your suggestion, we have combined the assessments for bias/heterogeneity/inconsistency into Section 3.2. (L317–L337) Accordingly, the figure numbers in the supplementary information portion have been changed appropriately.
Comment3: Assessment of heterogeneity - what about the EP+ICI vs EP studies?
Response3: Thank you for raising this very important point. Following the reviewer's comments, we assessed the heterogeneity for ICIs+EP vs. EP. One study each comparing Pem+EP vs. EP, Dur+EP vs. EP, and Atz+EP vs. EP (three studies in total) was included in this evaluation. The results are shown in Figure S3, where I2 = 0.0%, indicating that it is unlikely that heterogeneity between these three studies influenced the final conclusions. (L324–L332)
We are confident that our revised manuscript will be suitable for publication in Current Oncology, and we look forward to receiving your editorial decision.
Thank you for your consideration.
Sincerely,
Koichi Ando
Department of Medicine, Division of Respiratory Medicine and Allergology, Showa University School of Medicine
1-5-8 Hatanodai, Shinagawa-ku, Tokyo, 142-8666, Japan
Tel: +81-3-3784-8532
Fax: +81-3-3784-8742
Email: [email protected]

Reviewer 3 Report
Excellent Revisions
Author Response
Dear Reviewer 3
We sincerely appreciate the time and effort you have dedicated to provide valuable feedback on our manuscript.
Comment: Excellent Revisions
Response: We are very pleased to know that our manuscript titled "Comparative Efficacy and Safety of Immunotherapeutic Regimens with PD-1/PD-L1 Inhibitors for Previously Untreated Extensive-Stage Small-Cell Lung Cancer: A Systematic Review and Network Meta-Analysis” (Manuscript ID: curroncol-1087233) will be accepted for publication upon the completion of minor revisions. We are grateful to all three reviewers for the insightful comments they provided during the review process, and we have incorporated changes to reflect most of their suggestions. The changes in this round are highlighted in yellow below and within the manuscript.
We are confident that our revised manuscript will be suitable for publication in Current Oncology, and we look forward to receiving your editorial decision.
Thank you for your consideration.
Sincerely,
Koichi Ando
Department of Medicine, Division of Respiratory Medicine and Allergology, Showa University School of Medicine
1-5-8 Hatanodai, Shinagawa-ku, Tokyo, 142-8666, Japan
Tel: +81-3-3784-8532
Fax: +81-3-3784-8742
Email: [email protected]
